# Effect of Radiation Crosslinking and Surface Modification of Cellulose Fibers on Properties and Characterization of Biopolymer Composites

**DOI:** 10.3390/polym12123006

**Published:** 2020-12-16

**Authors:** Petr Lenfeld, Pavel Brdlík, Martin Borůvka, Luboš Běhálek, Jiří Habr

**Affiliations:** Faculty of Mechanical Engineering, Technical University of Liberec, Studentska 2, 461 17 Liberec, Czech Republic; pavel.brdlik@tul.cz (P.B.); martin.boruvka@tul.cz (M.B.); lubos.behalek@tul.cz (L.B.); jiri.habr@tul.cz (J.H.)

**Keywords:** biopolymer composite, polymer, biopolymer, natural fibers, surface modification, radiation crosslinking, mechanical behavior, fracture surface

## Abstract

Recently, polymers have become the fastest growing and most widely used material in a huge number of applications in almost all areas of industry. In addition to standard polymer composites with synthetic matrices, biopolymer composites based on PLA and PHB matrices filled with fibers of plant origin are now increasingly being used in selected advanced industrial applications. The article deals with the evaluation of the influence and effect of the type of surface modification of cellulose fibers using physical methods (low-temperature plasma and ozone application) and chemical methods (acetylation) on the final properties of biopolymer composites. In addition to the surface modification of natural fibers, an additional modification of biocomposite structural systems by radiation crosslinking using gamma radiation was also used. The components of the biopolymer composite were a matrix of PLA and PHBV and the filler was natural cellulose fibers in a constant percentage volume of 20%. Test specimens were made from compounds of prepared biopolymer structures, on which selected tests had been performed to evaluate the properties and mechanical characterization of biopolymer composites. Electron microscopy was used to evaluate the failure and characterization of fracture surfaces of biocomposites.

## 1. Introduction

Polymers and their composites are currently among the most progressively developing materials and there can be no doubt that they are key to modern technical society. In recent years, research into these materials has not only focused on highly durable synthetic composites, but also on renewable and sustainable natural materials [1]. This trend responds to the global problem of growing plastic waste, which is stimulated by the growing awareness of consumers, producers and governments. It is estimated that we are currently consuming oil at an unsustainable rate, and to alleviate these problems, governments in many countries are enacting laws that encourage the use of recycled, renewable and biodegradable polymers [2]. The new sustainable platform of the Institute for European Environmental Policy (IEEP) is Think 2030, which will direct EU member states towards the circular economy of plastic products [3].

The primary disadvantage of composite materials is the fact that two different components of the system make their reuse or recycling considerably complicated [4]. Another disadvantage, especially when using traditional reinforcements based on glass, carbon and aramid fibers, is the high energy intensity of their production [5]. In this respect, matrices based on thermoplastic polymers offer the advantage of being able to thermally further recycle and reuse these materials [6]. The most common form of recycling of composite materials are methods based on mechanical processing using various grinding methods, where the resulting products are subsequently reused in secondary applications. However, it should be noted that most composites end up mainly in landfills. At present, however, intensive research is also devoted to pyrolysis and solvolysis. Although these are very progressive technologies that are still evolving, their current technological state is considered to be either highly energy intensive or necessitates further chemical and recycling steps [7]. One of the possibilities in how to overcome the above-mentioned environmental shortcomings is to use sustainable composites based on renewable resources. These materials, often called “green” composites, are based on natural fillers and reinforcing elements that use biopolymer matrices from renewable sources [8]. The lower environmental impacts of these materials can be used mainly in applications that do not require excellent mechanical properties (secondary and tertiary structures or consumer goods such as packaging, covers, gardening supplies, etc.) [9].

The most studied biopolymer today is polylactic acid (PLA), which belongs to the family of linear aliphatic polyesters. It is a biocompatible and compostable (industrially biodegradable) thermoplastic polymer, which is obtained primarily by lactide ring opening polymerization. Lactide itself can be made from lactic acid, which is obtained by fermenting glucose or dextrose from renewable sources such as corn, sugar cane or sugar beet using various bacteria [10]. Although PLA exhibits interesting physical and mechanical properties, such as high tensile strength (50–70 MPa) and modulus of elasticity (3 GPa), excellent transparency, good process ability or biodegradability, its low impact strength and slow crystallization, which results in low temperature resistance, limit its use in more advanced technical applications [11]. The second most frequently studied group of biopolymers is polyhydroxyalkanoates (PHA), which also belongs to the group of linear polyesters. They are synthesized as intracellular carbon storage compounds and act as an energy reservoir for bacteria in the absence of food. PHAs are widely used in various biomedical applications due to their excellent biocompatibility and biodegradability [12].

The most important PHAs are poly (3-hydroxybutyrate) (PHB) and poly (3-hydroxybutyrate-co-3-hydroxyvalerate) (PHBV). PHB is a highly crystalline and brittle polymer with a high melting point and stiffness. To increase greater flexibility and improve process ability, PHB is copolymerized with 3-hydroxyvalerate (HV). The resulting PHBV is a copolymer of PHB with randomly arranged 3-hydroxybutyrate (HB) groups and 3-hydroxyvalatate (HV) groups [13]. The copolymer shows an increase in elongation at break and at the same time a decrease in the modulus of elasticity, crystallinity and melting point with increasing content of the HV group [14].

Today, it is possible to use natural material not only as composite matrices (biopolymers), but also as natural fibers, thus replacing synthetic fibers. The growing interest in natural fibers is mainly due to their good mechanical properties, low density and biodegradability at the end of their life cycle and low production costs. It follows from the above factors that the price for natural fibers is three times lower than for glass fibers, four times lower than for aramid fibers and five times lower than for carbon fibers [1]. Fibers of plant origin are cell walls found in the stems and leaves of plants. The fiber is an elementary linear structure, has a characteristic longitudinal and transverse cross-section and consists of primary chemical components [2]. The fibers are composed of cellulose, hemicellulose, lignins, aromatics, waxes and other lipids, ash and water-soluble compounds [3]. In this respect, plant natural fibers are an optimized structure, tested by evolution. Compared to glass or carbon fiber processing, natural fibers have a better impact on the environment and safety when using products with natural fibers. The disadvantages are flammability, variable quality and the need for drying.

The incorporation of natural reinforcing fibers into biopolymers can not only reduce the cost of the resulting composites, but also produce composites with a wide range of potential applications with many environmental benefits. The final properties of biocomposites reinforced with natural fibers are influenced by several factors, both from the point of view of fibers and from the point of view of the matrix. From the point of view of the fibers, the degree of filling, the length of the fibers, the morphology of the fibers, the orientation of the fibers and the distribution of the fibers have an effect. From the point of view of the biopolymer matrix, the resulting properties will depend on the molecular weight and the processing conditions. Another important parameter in green composites is the adhesion between the hydrophobic matrix and natural fibers, and the even distribution of the fibers [15]. Lignocellulosic fibers are generally hydrophilic and cellulose (microfibrils) is their main structural and reinforcing component [16]. Spirally wound cellulose microfibrils provide strength, stiffness and structural stability to natural fibers. The chemical structure of cellulose, which is responsible for its hydrophilic character, contains three reactive hydroxyl groups (OH). Two of them form hydrogen bonds inside cellulose macromolecules [17].

To improve the compatibility between the hydrophilic fibers and the hydrophobic matrix, various surface treatments using various chemicals have been developed over the years. However, these modifications usually use a large number of solvents [18]. The most commonly used chemical surface treatments for plant fibers include sialylation [19], acetylation [20] or maleic anhydride interfacial compatibilizers [21]. However, the handling and disposal of hazardous chemicals is not environmentally attractive. Alternatively, more environmentally friendly methods involving various physical or biological surface modifications of natural fibers have also been explored in recent years [22]. Among these methods, the plasma treatment of natural fibers is the most interesting from the point of view of industrial applicability, mainly due to short operating times of modification and low operating costs [23]. By plasma modification, various functional groups can be added to the surface of natural fibers, and these functional groups can form stronger covalent bonds with the matrix, leading to a better interfacial interface and increased mechanical properties. In systems using atmospheric plasma, the total plasma density is much higher, which increases the rate and rate of incorporation of ionized molecules into the surface of the material. The fibers then have a higher surface energy. When treating the surface of fibers with low-pressure plasma, there is no mechanical action and the thermal effect on the surface of the fibers is minimal [12]. Additionally, surface etching due to plasma treatment can improve the surface roughness and lead to a better connection to the matrix by means of mechanical locks [18]. The plasma treatment can generate a variety of surface modification effects depending on the ionized gas type, exposure duration, microwave strength and the distance of the fibers from the plasma source. A detailed study of the principles and application of plasma surface modification of natural fibers has been described by Sun and team [24]. In several publications [23,25,26] was, after usage plasma surface treatment, evaluated enhancement of surface adhesion of natural fibers with the PLA matrix, which evoke thermal stability, wettability and mechanical properties enhancing. The influence of dielectric barrier plasma discharge (DBD) on the resulting mechanical properties of biocomposite systems based on PLA/PALF has been investigated in our previous work [27]. A recent study then compared the effect of the content of technical cellulose fibers and their modification by DBD and O_3_ on the crystallization of PLA biocomposites [28]. Simultaneously, our colleagues recently investigated the possibility of modifying natural fibers by treating the surface with ozone [29], which is a process of oxidizing the surface of natural fibers with a reactive gas that acts as an intense oxidizing agent.

Another interesting and industrially applicable physical approach to improving interfacial adhesion and mechanical properties is the subsequent and additional exposure of manufactured composites to the effects of radiation crosslinking. Radiation crosslinking of polymers is based on the bombardment of molecules by a stream of high-energy electrons or gamma rays. This energy is absorbed by the material, radicals are formed (breakdown of C-H bonds), which gradually react with each other and form the desired connection. As a result of ionizing radiation, linking macromolecular chains emerge and form the spatial grid. The network is thus formed by the gradual joining of two free radicals between adjacent chains to form a C-C bond. The irradiation cross-linking of thermoplastic materials is performed separately after processing. Ionizing radiation includes accelerated electrons, gamma rays and X-rays. Gamma rays have a high penetration capacity. These are not only capable of converting monomeric and oligomer liquids, but can also cause major changes in the properties of solid polymers through cross-linking. The cross-linking level can be adjusted by the irradiation dose. From the results of previous studies, it is apparent, that the radiation crosslinking is a very efficient method for modifying the final properties of the polymers. However, some knowledge remains unexplained so far, and therefore each new finding about the effect of radiation crosslinking on the mechanical properties and behavioral characterization of polymer materials may contribute to a better understanding of the issue, especially in the field of biopolymer composites.

## 2. Materials and Methods

### 2.1. Materials 

Two biopolymer matrices were chosen for the preparation of biocomposites. The first polymer was PLA material designated Ingeo Biopolymer 3251D from NatureWorks LLC (Minnetonka, MN, USA). It is a biopolymer intended for processing by injection molding technology and is standardly supplied in the form of granules. The second polymer was PHBV material Enmat Y1000P from TianAn Biopolymer (China). It is a biopolymer, which, like the PLA material, is supplied in granules and is intended for processing by injection, extrusion, blow molding and thermoforming technologies. The natural filler was technical cellulose (CeF) purchased from Arbocel ZZC 500, J. Rettenmaier & Söhne (Germany). Producer of technical cellulose declare the average fibers size as a 400 μm, thickness 45 μm and chemical composition: 80-90 mass% cellulose and 10–20 mass% calcium carbonate [28]. Microscopic images of the cellulose fibers are shown in Figure 1.

Modification of biocomposite systems (matrix + CeF) and physical and chemical modification of the surface of cellulose fibers were used to influence the final properties of biocomposite materials. The radiation crosslinking method on the BGS Beta-Gamma-Service GmbH and Co. (Germany) plant was used to modify biopolymer composite systems with a PLA and PHBV matrix with two different doses of gamma radiation 50 and 100 kGy (signification R). Both physical (plasma and ozone) and chemical processes (acetylation) were used for surface modification of cellulose fibers. The MSV Systems CZ device was used for plasma modification (signification P) of cellulose fibers, enabling low-temperature plasma technology, consisting of two parallel electrodes covered with a 1 mm layer of dialectic, between which a volumetric cold plasma discharge burned in the filament mode. The electrodes were rectangular in size 50 mm × 60 mm with a thickness of 8 mm without active cooling and the distance between the electrodes was 15 mm. Plasma modification of cellulose fibers (see Figure 2a) was performed under the following conditions: voltage 20 kV, frequency 3 to 10 kHz, and power 200 W. From the acetylation point of view (signification A), the cellulose fibers were soaked in a reaction bath having the following composition: 85 g of acetic anhydride, 1 g of sulfuric acid, and 914 g of concentrated acetic acid. The cellulose fibers were soaked in a bath in a closed vessel for 24 h at a bath temperature of 20 °C. Subsequently, rinsing was performed under running water until neutral pH and finally neutralization was performed with sodium carbonate with a concentration of 1 g∙L^−1^ (see Figure 2c). The GO 5LAB (Triotech, Czech) apparatus was applied to generate ozone (signification O) during surface treatments. Natural fillers (CeF) were exposed to dried airflow (dew point +5 °C, flow rate 3 L min^−1^) with a concentration of oxygen nearly 100% and generation of ozone by 20 mg L^−1^. Ozone concentration was measured by the UV photometer LF200 (Greisinger, Germany). Elimination of the redundant ozone was ensured in ozone destruction units, which is filled with active carbon. Treating time was 4 h. Additionally, the natural fillers were cleaned in pure airflow for 1 h to replace the rest of the ozone (see Figure 2b). 

### 2.2. Preparation of Biopolymer Composites 

For suitable process adjustment, the decomposition temperature of natural filler and its amount of inorganic phase were evaluated before composite production. Thermal degradation has been performed using thermal gravimetric analysis (TGA) on a TGA2 instrument (Mettler Toledo, Greifensee, Switzerland). Samples were heated from 50 to 600 °C under N_2_ atmosphere and further from 600 to 800 °C under O_2_ atmosphere at the heating ramp of 10 °C min^-1^. The initial decomposition temperature was determined at 5% mass loss. Decomposition of cellulose fibers at 263 °C. The residual mass was evaluated at 800 °C. The evaluated average amount of the inorganic phase (ash) was 13 mass% in cellulose fibers (see Figure 3a). 

The chemical composition of the ash was determined by Fourier transform infrared spectroscopy (FTIR), at ambient temperature (23 °C). FTIR spectra of cellulose fibers and their ashes were recorded using a Nicolet iS10 FTIR spectrometer (Nicolet, Rhinelander, WI, USA) with DLaTGS (deuterated lanthanum α alanine doped triglycine sulfate) detector and diamond attenuated total reflection (ATR). In all cases, a total of 64 scans at a resolution of 4 cm^−1^ were used to record the spectra. The spectra were taken over a wavenumber range of 400–4000 cm^−1^ and were obtained with respect to a background, which was taken of the air previously and under the same measurement conditions (see Figure 3b). From the FTIR spectra, it can be stated that the ash of the cellulose fibers to calcium carbonate. On the FTIR spectrum of calcium carbonate the fundamental bands can be seen at: 708 cm^−1^ (in-plane bend), 883 cm^−1^ (out-of-plane bend) and at about 1400–1500 cm^−1^ (asymmetric stretch) [30]. The ash of technical cellulose fibers contained the same bands at wavenumbers 875 cm^−1^ and 1416 cm^−1^. The results of ash chemical composition correspond to material composition of Arbocel ZZC 500.

Before the processing was polymer PLA and polymer PHBV dried in Maguire Low Pressure Dryer (LPD 100) under the following conditions: temperature 80 °C, time 120 min and vacuum 0.8 bar to a residual moisture content of 0.025%. Prior to compounding, the natural fibers were dried in a Venticell 222 hot air oven with forced air circulation at a temperature of 80 °C for 120 min. Biocomposite pellets were prepared by twin screw extruder (ZAMAK EHP-2x130di) followed by water bath and pelletizer. Temperature profile of the extrusion line was set from 160 up to 190 °C for the polymer PLA and PHBV. Cellulose fibers were dosed directly into the melting chamber of extruder in the recommended front position by the external device, working on the gravimetric principle. The reason for dosing in the front parts of the extruder is to prevent excessive shear stress of cellulose fibers during melt compounding and thus their damage or thermal degradation. Pelletized compounds PLA and PHBV passed through a water bath and thus were before injection molding dried at the Maguire Low Pressure Dryer (LPD 100) under following conditions: temperature 80 °C, time 180 min and vacuum 0.8 bar. A total of 12 biocomposite materials were compared for the same percentage volume of natural fillers in the matrix (20 wt% cellulose fibers) with different type of matrix (PLA and PHBV), different type of surface modification of fibers (plasma modification, ozonation and acetylation) and different value of gamma radiation (50 and 100 kGy), which were compared with a pure unfilled biopolymer matrix of PLA and PHBV (see Table 1). Testing specimens were injection molded according to ISO 527 and ISO 178 on the injection molding machine (ARBURG 270S 400-100) with increasing temperature profile (165–190 °C for the polymer PLA and 160–180 °C for the polymer PHBV) of the melting chamber and injection speed 35 cm^3^·s^−1^ for PLA and injection speed 25 cm^3^·s^−1^ pro PHBV. Resulted biocomposites and control samples are depicted in Table 1.

### 2.3. Methods

#### 2.3.1. Uniaxial Tensile Testing

For tensile testing of injection molded specimens a Tiratest 2300 were used. The measurement was performed according to the STN EN ISO 527 standards with the tested specimen 1A. The specimens were strained at room temperature up to break at a test speed 5 mm·min^−1^. The module of tensile elasticity (Young’s modulus) was determined at a reduced test speed 1 mm·min^−1^. From the stress–strain dependences, tensile strength was calculated. The measurement was performed on 10 test specimens at a temperature of 23 °C. The mean value and standard deviation were calculated. The conditioning took place in conditions according to STN EN ISO 291. The LabNet program was used for the evaluation.

#### 2.3.2. Flexural Testing

The bending test (according to STN EN ISO 178) was measured on a Hounsfield H10KT tearing machine. The conditioning was performed according to the STN EN ISO 291 standard. Test specimens measuring 80 mm × 10 mm × 4 mm were placed on two supports. To determine the modulus of elasticity, the measurement was performed at a speed of 2 mm·min^−1^ with preload according to the standard. Measurements were made on 10 test specimens at 23 °C and the mean and standard deviation were calculated. The measured values were recorded by the Qmat program. For the secant modulus, it was necessary to determine the stress (*σ_f_*_2_ and *σ_f_*_1_) at a relative elongation of 0.05% (*ε_f_*_1_) and 0.25% (*ε_f_*_2_), respectively. The modulus of elasticity itself was then calculated based on Equation (1): (1)$$E_{f}=\frac{\sigma_{f2}-\sigma_{f1}}{\varepsilon_{f2}-\varepsilon_{f1}}\;$$
where *E_f_* is the secant modulus, *σ_f_*_2_ is the stress at relative elongation 0.25%, *σ_f_*_1_ is the stress at relative elongation 0.05%, *ε_f_*_2_ is the relative elongation 0.25% and *ε_f_*_1_ is the relative elongation 0.05%.

#### 2.3.3. Charpy Impact

The Charpy method according to the STN EN ISO 179-1 standard on the Resil 5.5 device was used to determine the impact strength. Prior to the actual measurement, the instrument was calibrated for mechanical resistance and air resistance at “idle” start-up. The conditioning was performed according to the STN EN ISO 291. The test specimens had dimensions of 80 mm × 10 mm × 4 mm. The measurement was performed on 10 test specimens at a temperature of 23 °C. After the measurement, the impact toughness a_cU_ was calculated and the mean value and standard deviation were calculated from the values.

#### 2.3.4. Scanning Electron Microscopy (SEM)

Scanning electron microscopy was used to determine a fracture surface analysis in the central part of test specimens of biocomposites with natural cellulose fibers without modification of the fiber surface, with the modification of the fiber surface and for samples after radiation crosslinking. The scanning electron microscope (SEM) was conducted on Carl Zeiss ULTRA Plus (Carl Zeiss, Oberkochen, Germany) at an acceleration voltage range of 2–5 kV. The surface of all fracture surfaces was provided with a thin layer of platinum (3 nm) to improve the conductivity of the samples.

## 3. Results and Discussion

Polymer composites with a biopolymer matrix are increasingly used in practice, and therefore it is very important to have knowledge about composite structures in relation to the possibilities of additional modification of biocomposites as a whole, or in relation to surface modification of the natural filler. The presence of fibers in biopolymer composites and the application of surface modifications of the filler or the prepared composite as a whole should have an impact on the properties and mechanical characterization of composite structures and the behavior of composites under the load.

### 3.1. Tensile Properties

The tensile properties of PLA and PHBV biocomposites are summarized in Figure 4. Figure 4 show not only the negligible effect of adding 20% cellulose fibers to the biopolymer matrix, but also the negligible effect of surface modification of natural cellulose fibers in both biocomposite systems with PLA and PHBV matrix on the tensile modulus. The SEM images show that the adhesion between the natural fibers and the biopolymer matrix at the interfacial interface was less satisfactory, the fibers were not perfectly surrounded by the polymer (see Figure 5). For biocomposite structures with PLA matrix and cellulose fibers without modification and with surface modification of fibers by physical or chemical means (see Figure 4) it was evident that the tensile modulus was almost comparable (differences are in the order of percentage units) and was comparable with the unfilled PLA matrix. Surface modification of cellulose fibers (except plasma treatment) did not affect the increase of the tensile modulus value compared to PHBV matrix or PHBV matrix with unmodified cellulose fibers and the lowest values were reached by PHBV biocomposite with modification of cellulose fibers by acetylation.

Furthermore, the results show a negative impact of additional radiation crosslinking on the tensile modulus of elasticity of the PLA matrix, which was greater the greater the value of radiation. In contrast, for the PHBV matrix, the use of radiation crosslinking had the opposite effect, the value of the tensile modulus increased (about 15%), although as with PLA, the increasing radiation value decreased the tensile modulus. In the case of the application of radiation crosslinking in PLA composites, the values were comparable both with the unfilled PLA matrix and with biocomposites without and with surface modification of cellulose fibers. In the case of application of twice the radiation value (100 kGy) during radiation crosslinking, the tensile modulus of PLA composite structures reached the lowest value. During radiation crosslinking, the dominant phenomenon of PLA and PHBV biomaterials was the cleavage of ester groups into mainly hydroxyl groups [31]. It was clear from the experimental measurements that with increasing radiation levels, the modulus of elasticity decreased for biocomposites based on the PLA matrix. The degree of crystallinity (Xc) of the biopolymer composite PLA increased after the addition of 20 wt% of CeF fibers. For the neat PLA matrix, the degree of crystallinity was 10.3%, for PLA composites, the Xc increased approximately by 5%. Additional radiation crosslinking of PLA composites did not increase the crystallinity degree. Radiation crosslinking caused the scission of internal macromolecular bonds and formation of free radicals. These subsequently reacted and formed a three-dimensional structure by joining two free radicals between adjacent chains. Crosslinking reactions in PLA polymers mainly affected amorphous regions or less crystalline regions in the structure, which as reflected in a decrease in mechanical properties. These results correspond to the results of the work of S. Tiptikorn and the team [32]. The different crystallinity of PLA and PHBV biocomposites resulted in non-identical effect of gamma radiation on the tensile modulus of elasticity (see Table 2 and Table 3). The mentioned cleavage of “took place” chains occurs mainly in irradiated amorphous regions [33]. Therefore, it can be assumed that a biopolymer with a higher content of the crystalline phase (see Table 3) was characterized by a higher resistance to radiation and thus a lower decrease in mechanical properties [34]. However, scissions in the linked molecules are the ones that causes the highest losses in mechanical properties to a crystalline polymer [35].

The increase in the tensile modulus of the PHBV biocomposite can be explained by the fact that the cleavage of the amorphous regions was eliminated by increasing the crystal structure of the PHBV biocomposite (see Table 3) after application of gamma radiation. The radiation probably caused the uncontrolled decay of large crystals, which subsequently served as nucleation nuclei for the formation of new crystalline formations [34]. Evidence of the influence of gamma radiation on the change of the crystal structure and on the modulus of elasticity in tension is also given in the studies of L. M. Oliver and colleagues [34,35], where the modulus of elasticity in tension remained almost unchanged even at 100 kGy.

From the results of tensile stress at the yield point (see Figure 6) it can be stated that for biocomposite structures with PLA matrix and cellulose fibers without modification and with surface modification of fibers by the physical or chemical way the values decreased to 65–70% of the yield stress value for a pure PLA matrix. The effect of the type of surface modification of the cellulose fibers was comparable to each other between the methods used. The effect of the surface treatment of cellulose fibers led to a slight increase in the value of tension compared to unmodified fibers. This may be due to the length of the fibers (the cellulose fibers used were short), the agglomeration of the fibers and also the imperfect adhesion of the fibers to the polymer matrix. Theoretically, continuous fibers would be able to have a significant positive effect, which would be able to transfer the inserted load. However, the situation was significantly different for the PHBV matrix, because for biocomposites without modification of cellulose fibers and with surface modification of fibers by physical or chemical means, the yield stress values were comparable to the pure PHBV matrix. The effect of any surface modification of cellulosic fibers used improved the yield stress values over non-surface modified fibers. After the application of radiation crosslinking to PLA biocomposites, the reduction of the yield strength value was even more pronounced, and by increasing the value of the radiation magnitude, the stress was halved. In PLA, there was a significant degradation of the polymer matrix due to its chemical composition. The higher the energy supplied, the deteriorating the properties, the more likely the chains are shortened or the molecular weight is reduced. The die then does not withstand tensile loading because it had become brittle. After the application of radiation crosslinking in the PLA biocomposite, the decrease compared to the pure PLA matrix was 65% at a radiation value of 50 kGy and more than 80% at a radiation value of 100 kGy. If we compared the biocomposite structure of PLA with 20% of cellulose fibers without radiation crosslinking and with radiation crosslinking, then the loss of stress value was 50% for a radiation value of 50 kGy and 75% for a radiation value of 100 kGy. Increasing the value of radiation from 50 kGy to the value of radiation 100 kGy during radiation crosslinking will reduce the value of tensile stress at the yield point by half. With higher levels of radiation, there was a higher level of chain cleavage and consequent decrease in molecular weight, decrease in glass transition temperature and decrease in mechanical and other properties [36]. The same dependence was detected in the work of L. Meihua and his team [37], who studied the effect of radiation crosslinking of PLA polymer containing “ballast fiber”. Additionally, in the works of S. Dadbin and his team [38,39] there was a significant decrease in “tensile strength” with increasing radiation levels.

For PHBV composites, the stress values for the low radiation value were slightly lower compared to the modified fibers and further decreased with increasing radiation value. However, the decrease in yield stress values was much smaller than for PLA biocomposites. PHBV did not appear to have such significant degradation of the polymer matrix due to its chemical composition. When applying radiation crosslinking of a biocomposite with a PHBV matrix, the decrease in the voltage value compared to a pure PHBV biopolymer matrix was 15% for a radiation value of 50 kGy and 50% for a radiation value of 100 kGy. If, as with the PLA matrix, we compared the biocomposite structure of PHBV with 20% of cellulose fibers without radiation crosslinking and with radiation crosslinking, then the yield stress value was reduced by 5% at a radiation value of 50 kGy, respectively by more than 40% at a radiation value of 100 kGy. By applying radiation crosslinking and doubling the value of the radiation, it led to a decrease in the tensile stress at the yield point by 40%. From the point of view of the matrix used (PLA or PHBV) it was clear from the results that the application of radiation crosslinking of biocomposite structures was a huge loss of properties for biocomposites with the PLA matrix (50 and 75%), but with the PHBV matrix it was much less negative (5% and 40%). This fact can be explained by the different chemical composition and the influence of gamma radiation on the structure [33]. The results show a decrease in properties, which was more pronounced in comparison with the level of modulus of elasticity in tension. The obtained results are in accordance with the results reported in the works of L.M. Oliver and team [34,35] and the study of S. Luo and team [33], which showed a significant decrease in “tensile strength“ when applying gamma radiation.

From the results of the nominal strain at the yield point (see Figure 7), it can be concluded that using cellulose fiber surface modification methods, the nominal strain value of biocomposite structures with a PLA matrix was reduced by about 50% compared to a pure PLA matrix. In the case of biocomposite systems with a PHBV matrix, the values of the relative elongation were approximately comparable, they only decreased with the application of radiation crosslinking with a high value of radiation. When using radiation crosslinking in PLA biocomposite structures, the decrease in the value of the relative elongation of 70% relative to the PLA matrix, respectively 40% compared to PLA composites without and with surface modification of natural cellulose fibers. For a radiation size of 100 kGy, the decrease was 80% respectively 65%. For PHBV biocomposite structures, the changes in values were about half. For a radiation value of 50 kGy, the decrease was 25%, resp. 10% and for a radiation value of 100 kGy there was a decrease of 55%, resp. 45%. Based on the values found, it can be stated that the addition of cellulose fibers decreased the ductility of the biocomposite system (especially for PLA matrix) and with the use of radiation crosslinking and with increasing gamma radiation the relative elongation decreased significantly. Increasing the radiation value from 50 to 100 kGy led to a decrease in ductility by almost 50% for both biopolymer matrices. Elongation decreased due to the fibrous filler, because due to their morphology they could not transfer the inserted load. The ductility of the matrix was lower after radiation crosslinking, as the structure was degraded due to chain shortening, which was caused by the energy supplied to the system.

### 3.2. Flexural Properties

Based on the flexural test performed it can be concluded that the flexural modulus (see Figure 8) did not substantially increase with the addition of cellulose fibers and fiber modifications to the PLA biopolymer matrix. A significant increase in the flexural modulus occurred only with the PHBV biopolymer matrix. This applies both to the application of radiation crosslinking and to the application of surface modification of cellulose fibers in both physical and chemical ways. This conclusion was confirmed by the fracture surfaces of biocomposite structures, which show significantly improved adhesion at the interfacial interface of the PHBV biopolymer matrix and natural cellulose fibers.

The flexural modulus values (see Figure 8) for the PLA biopolymer matrix show that the flexural modulus was comparable for almost all samples (pure PLA, unmodified cellulose fibers, surface modified cellulose fibers and radiation crosslinking). Adhesion at the interfacial interface in PLA composites was insufficient (see Figure 9). The modulus of elasticity generally did not change substantially when using fiber fillers without good interfacial adhesion.

In contrast, in the case of a biocomposite structure with a PHBV matrix, the flexural modulus increased by about 15% after application of surface modifications of the fibers, where the adhesion was significantly better (see Figure 10). While in the biocomposite structure of PHBV without modification of the surface of the cellulose fibers the fracture surface was not homogeneous and dark spots were visible, cavities after fiber extraction, adhesion had improved after surface modification, the matrix coated the fiber reinforcement and cellulose fibers were broken. After the application of radiation crosslinking, the flexural modulus increased by a quarter, 25% compared to unmodified cellulosic fibers. This was despite the fact that the adhesion at the interface between the matrix and the fibers was minimally affected, but rather the bonds in the polymer matrix were affected, which could occur due to the energy supply to the system and change of morphological system (increase in molecular weight). The effect of the increased radiation value had a negligible effect on the change in the flexural modulus.

The flexural strength (see Figure 11) of PLA and PHBV biocomposite structures shows similar dependences as the yield strength. In the case of PLA biocomposite with uncoated fibers and surface-modified fibers, the values obtained were always lower than in the case of unfilled PLA; the values decreased to about 80% of the stress value of the pure PLA matrix. The effect of the type of surface modification of the cellulose fibers was comparable to each other between the methods used. Short cellulose fibers cannot sufficiently contribute to increasing the flexural or tensile strength limit if interfacial adhesion is not sufficient. Additionally, after the application of radiation crosslinking, the values of the flexural strength were even lower. The matrix was degraded by radiation crosslinking or by supplying energy to the system. When using radiation crosslinking of a PLA biocomposite, the decrease compared to a pure PLA matrix was 60% for a radiation value of 50 kGy and 80% for a radiation value of 100 kGy. If we compared the biocomposite structure of PLA with 20% of cellulose fibers without radiation crosslinking and with radiation crosslinking, the reduction in stress was 50% for 50 kGy and 75% for 100 kGy, which was half the stress for half the radiation (50 kGy). The reason was the considerable cleavage of the PLA biopolymer chains, which was even more pronounced with increasing levels of gamma radiation. A similar dependence was detected in the work of L. Meihu [37] and in the work of Dadbin and his team [38,39]. 

In the case of biocomposite systems with a PHBV matrix, the flexural strength values were approximately comparable both in the case of biocomposites with cellulose fibers without modification and in the case of fibers with surface modification by physical or chemical means and began to decrease with increasing radiation value. When applying radiation crosslinking of a biocomposite with a PHBV matrix, the decrease in stress value compared to a pure biopolymer matrix PHBV or a matrix filled with 20% unmodified cellulose fibers was about 15% at a radiation value of 50 kGy and 50% at a radiation value of 100 kGy. Modification of a biocomposite system with a PLA matrix by radiation led to the destruction of biopolymer bonds and thus to a significant decrease in values. In the case of the PHBV matrix, this occurred only at a high value of gamma radiation (100 kGy). These results confirm the previous conclusions that in the case of PHBV, there was no fundamental and undesired destruction of the polymer structure after radiation crosslinking if the supplied energy was low. The decrease in the flexural strength was not so great due to the higher content of the crystalline structure of PHBV biocomposites and the influence of gamma radiation on the formed structure. This result corresponds to the results of studies by L. M. Olivera [34,35]. At higher levels of gamma radiation, due to the considerable chain splitting, there is also a significant decrease in mechanical properties of the PHBV biocomposite. The negative effect of higher levels of gamma radiation on the decrease of mechanical properties of PHBV composite systems was recorded in the work of K. Iggui [40].

### 3.3. Charpy Impact Properties

The impact strength of both PLA and PHBV matrix was relatively low and the addition of natural cellulose fibers as a filler to the composite system further reduced this property. From the measured values (see Figure 12) it was evident that the addition of fibrous filler significantly reduced the impact strength from the point of view of the biocomposite structure as a whole. The results of impact measurements for the biocomposite structure with PLA matrix at a temperature of +23 °C show a significant loss of toughness by the addition of cellulose fibers, which was not affected by the method of surface modification of the natural filler. The decrease in impact strength was almost 50%. In the case of a biocomposite with a PHBV matrix, the effect of the added cellulose fibers was also negative and the impact strength was reduced by an average of 20% compared to the unfilled PHBV matrix.

Due to the application of radiation crosslinking of the PLA composite, the impact toughness was further reduced: at a radiation value of 50 kGy the impact toughness was only 40% and for a radiation value of 100 kGy was only 10% of the impact toughness value relative to the unfilled PLA matrix. The decrease in impact strength when the radiation value was doubled, from 50 to 100 kGy, was then 65%. The application of radiation crosslinking to a biocomposite with a PHBV matrix reduced the impact strength to 50% (at a radiation value of 50 kGy), respectively 40% (at a radiation value of 100 kGy) compared to an unfilled PHBV matrix. Thus, the application of radiation crosslinking in PLA and PHBV composites led to a further decrease in impact strength values, as the embrittlement of the matrix occurred. This conclusion corresponds with the published results [33,35], where an obvious decrease in impact strength was observed during the application of gamma radiation in PHBV biocomposites.

### 3.4. Characterization of Failure and Microscopy of Fracture Surfaces

An important aspect of fibrous filler composites in terms of their application is not only the quality of interfacial adhesion between the fibers and the polymer matrix, which is a decisive factor in the use of fiber reinforcement potential (especially those strength–glass, carbon, etc., which find application in synthetic polymers), but also the evaluation of the fracture surface after modifications of the fiber surface and also after radiation crosslinking. Another important aspect of composite materials is the placement of the filler in the matrix, as the agglomeration of the filler causes inhomogeneity that degrade the reinforcing effect of the fibrous filler. SEM images of fracture surfaces of PLA and PHBV biopolymer composites with cellulose fibers without surface modification, with physical and chemical surface modification and also after radiation crosslinking were taken for evaluation of fracture surfaces and evaluation of interfacial adhesion at the interface of plant fibers and the biopolymer matrix. For the evaluation of fracture surfaces, Figure 13 shows images of fracture surfaces of PLA biocomposite structures without modification, Figure 14, Figure 15 and Figure 16 show fracture surfaces of PLA biocomposites with cellulose fiber modification and Figure 17 images of fracture surfaces of PLA biopolymer composites after gamma radiation. Figure 18 shows images of fracture surfaces of biocomposite structures with the PHBV matrix without modification, Figure 19, Figure 20 and Figure 21 show images of fracture surfaces of PHBV biocomposites with cellulose fiber modification and Figure 22 shows fracture surfaces of PHBV biopolymer composites after gamma radiation.

Microscopic images of the fracture surfaces of PLA biocomposites (see Figure 13, Figure 14, Figure 15 and Figure 16) show that the fracture surface was not homogeneous and contained both clusters of fibers of hundreds of micrometers and individual fibers, both for cellulose fibers without surface modification, and surface-modified fibers. The fiber clusters were difficult to saturate and thus did not improve the mechanical properties that are standard in glass or carbon composites. The fiber clumps were then the weak points of the biocomposite where the failure occurred. After the application of surface modifications, the adhesion at the interfacial interface was improved, the matrix material partially enveloped the fiber reinforcement, but it was still not sufficient to increase the values of mechanical properties. The cellulose fibers themselves were torn, not drawn. The application of gamma radiation (see Figure 17) changed the character of the fracture, the character of the matrix changed. The brittle character of the fracture could be inferred from the fracture surface and it could also be seen that the adhesion to the fibers was very low. 

Microscopic images of fracture surfaces of PHBV biocomposites (see Figure 18, Figure 19, Figure 20 and Figure 21) show, similarly to PLA biocomposites, that the fracture surface was not homogeneous and contained both fiber clusters and individual fibers, both for cellulose fibers without surface modification and for surface-modified fibers. Clusters of fibers were critical places where fracture occurred. Additionally, as in the previous case, the fiber clusters were difficult to saturate. After the application of surface modifications, there was no significant improvement in adhesion at the interfacial interface. The application of gamma radiation (see Figure 22) to PHBV biocomposites changed the nature of the refraction (especially with a high value of gamma radiation of 100 kGy), the fracture surface shows a brittle nature of the refraction. When applying gamma radiation of 50 kGy, it is a tough refraction, but the nature of the refraction was different compared to the matrix without irradiation.

## 4. Conclusions

In recent years, biopolymer composites have become the focus of a wide range of research activities, material manufacturers, processors and end users. Biocomposite materials based on PLA and PHB matrices and plant fibers have great potential for wide application use and are beginning to gain in importance. In the case of biocomposites, two important sectors can be distinguished, in which products are used in technical applications and in medical applications. The use of biocomposites and natural fibers in technical applications offers wide possibilities for technical innovation and sustainable development. Today, they are no longer just biodegradable packaging, foils and disposable products, but recently, there have been a significant demand for these composites for various technical products, such as automotive parts, furniture components, design products and electrical engineering, but also in the aerospace industry. Polymer composites with the biopolymer matrix and natural fibers have increasingly been used mainly in interior applications in terms of environmental requirements and environmental sustainability.

Experimental results obtained from the evaluation of mechanical parameters concerning prepared biocomposite systems were related to the chemical composition of used matrices, type and shape of filler-cellulose fibers, percentage of fibers, fiber surface treatments, fiber adhesion to polymer matrices, homogeneity of fiber distribution in polymer matrix, fracture character and also radiation that was used to crosslink selected samples.

The application of surface modifications to cellulose fibers did not achieve the expected effects known from other commercially available fibers. There were no major morphological changes at the fiber surface level that would have a determining effect on the adhesion of the fibers to the polymers. There appeared to have been chemical changes (opening of surface bonds leading to a change in hydrophilicity/hydrophobicity), which, however, were insufficient to significantly affect and improve adhesion at the fiber–polymer interface, especially in the PLA matrix. Based on the achieved results, with some exceptions, there were no positive changes in terms of increasing mechanical parameters. At the same time, conclusions from other publications and studies were confirmed.

The chemical composition of the used PLA and PHBV matrices, which was different, had a significant effect not only on the adhesion at the interfacial interface, but also on the final properties after radiation crosslinking. In terms of mechanical properties, better values of mechanical properties were achieved for the PHBV matrix both in adhesion and the effects of gamma radiation. The results were confirmed and at the same time, expanded knowledge in this area.

The addition of 20% cellulose fibers did not significantly improve the evaluated mechanical parameters. This fact can be explained by a number of factors and their combination. Cellulose fibers of the above dimensions were not straight, so they were easily intertwined, which was initiated by a change in surface tension for the modifications made and, last but not least, by the method of preparation (temperature–pressure processes) of test specimens. As a result of the formation of cellulose agglomerates, there was no uniform dispersion, the cellulose agglomerates could then function as weak points, especially if they were not coated with a polymer matrix where the breakdown occurred. These effects were also observed in other experimental work.

During the process of radiation crosslinking, bonds were broken down and free radicals were formed. Following the formation of free radicals, other processes might occur, such as cleavage of macromolecular chains or interconnection of macromolecular chains, or both. Which process will predominate will be related to the chemical composition of the polymer, the ratio of amorphous and semicrystalline regions, or the presence of the filler and its chemical composition, the dose of radiation used and the process conditions. Polymers that contain an oxygen atom in the backbone are more sensitive to radiation in the negative sense of the word. Cellulose is also sensitive to the effects of radiation. Based on the chemical composition, it can therefore be assumed that the polymer crosslinks will degrade or shorten during radiation crosslinking. These theoretical assumptions, which were also found in the above studies, were confirmed by measurement in the PLA matrix. On the contrary, in the case of the PHBV matrix, radiation crosslinking had a rather positive impact and negative effects occurred only due to high radiation values.

The acquired knowledge about biopolymer composites, their structures, constructions and modifications, including knowledge about the characterization of mechanical properties, behavior under load and fracture behavior is important both in terms of advanced application possibilities but also in terms of biopolymer composites as progressive materials with a wide range of environmental benefits.

## Figures and Tables

**Figure 1 polymers-12-03006-f001:**
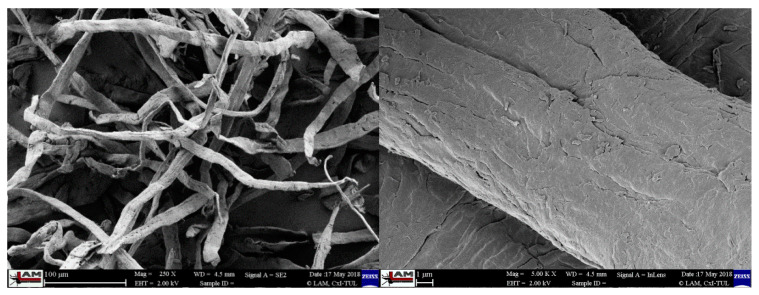
Microscopic image of cellulose fibers.

**Figure 2 polymers-12-03006-f002:**
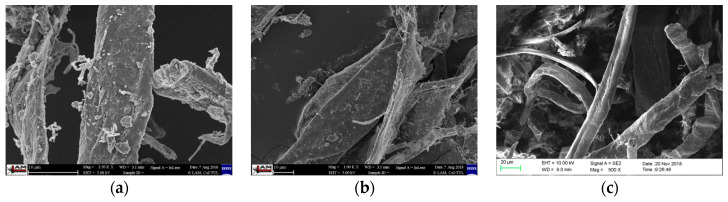
Microscopic image of cellulose fibers (CeF) after plasma modification (**a**), after ozonation (**b**) and acetylation (**c**), SEM.

**Figure 3 polymers-12-03006-f003:**
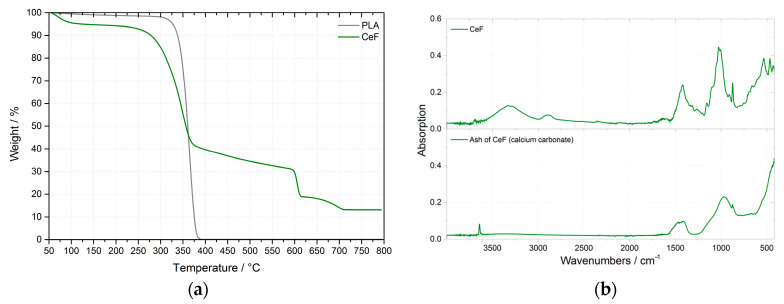
TG curves of PLA and cellulose fibers (CeF) decomposition (**a**) and FTIR spectra of cellulose fibers (CeF) before and after thermal decomposition (**b**).

**Figure 4 polymers-12-03006-f004:**
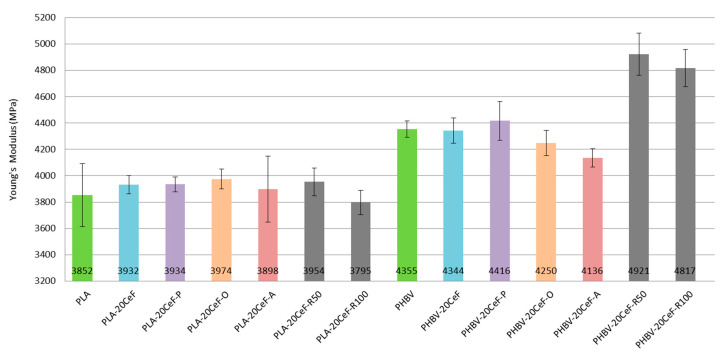
Tensile modulus of elasticity (Young’s modulus) of PLA and PHBV biopolymer composites.

**Figure 5 polymers-12-03006-f005:**
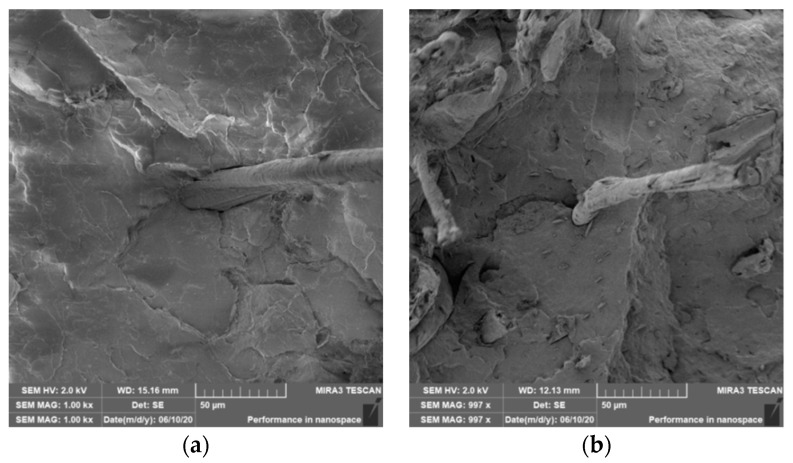
Microscopic image of fracture surfaces of the biocomposite structure with the PLA matrix and cellulose fibers (CeFs) after application of ozonation (**a**) and the PHBV matrix after plasma application (**b**), SEM.

**Figure 6 polymers-12-03006-f006:**
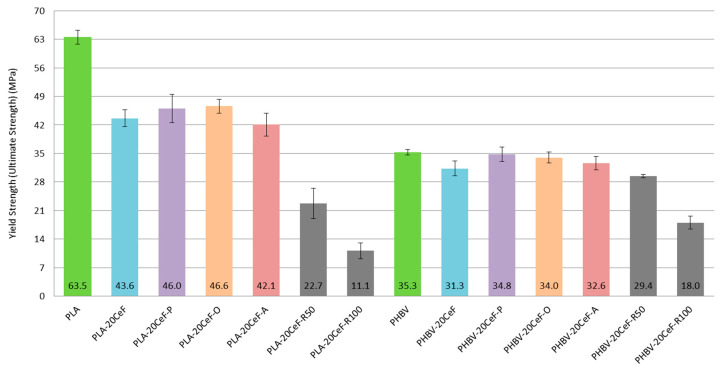
Yield strength (ultimate strength) of PLA and PHBV biopolymer composites.

**Figure 7 polymers-12-03006-f007:**
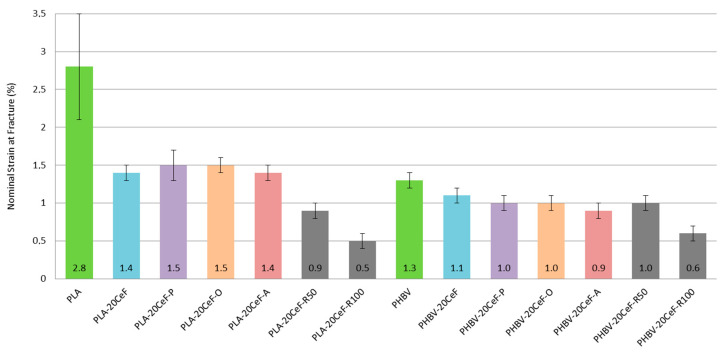
Nominal strain at fracture of PLA and PHBV biopolymer composites.

**Figure 8 polymers-12-03006-f008:**
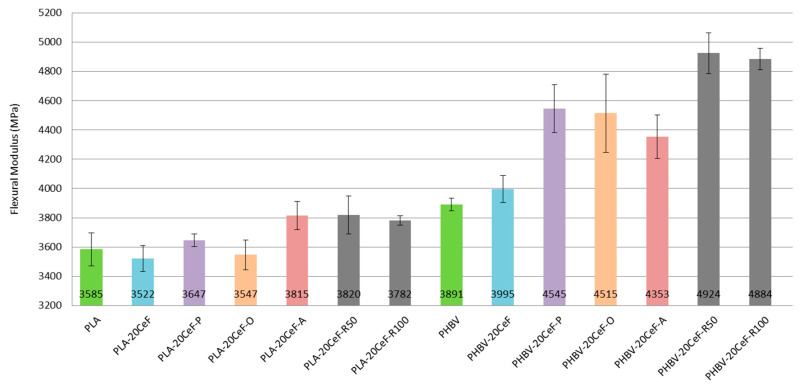
Flexural modulus of PLA and PHBV biopolymer composites.

**Figure 9 polymers-12-03006-f009:**
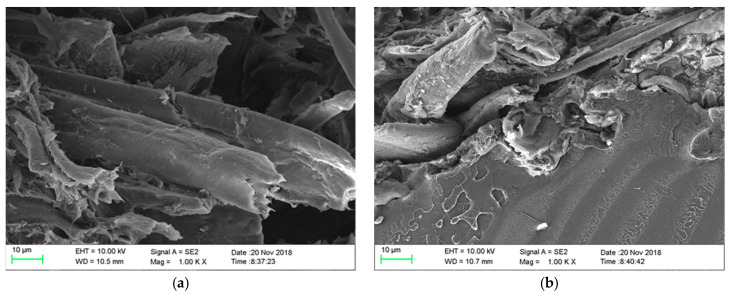
Microscopic image of fracture surfaces of biocomposite structure PLA with cellulose fibers (CeF) with acetylation (**a**) and after radiation crosslinking 50 kGy (**b**), SEM.

**Figure 10 polymers-12-03006-f010:**
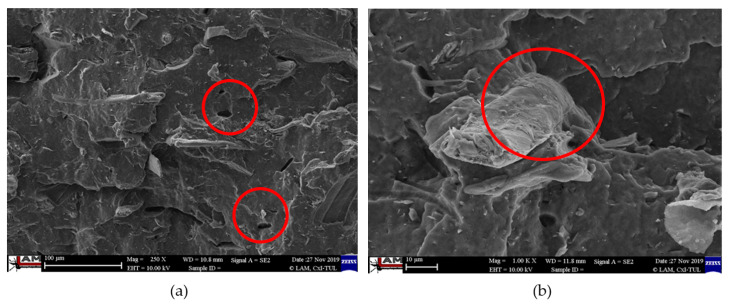
Microscopic image of fracture surfaces of the PHBV biocomposite structure with cellulose fibers (CeF) without surface modification (**a**) and after ozonation (**b**), SEM.

**Figure 11 polymers-12-03006-f011:**
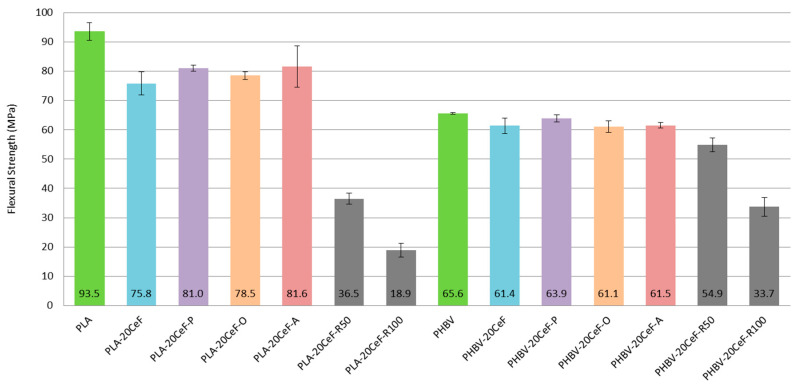
Flexural strength of PLA and PHBV biopolymer composites.

**Figure 12 polymers-12-03006-f012:**
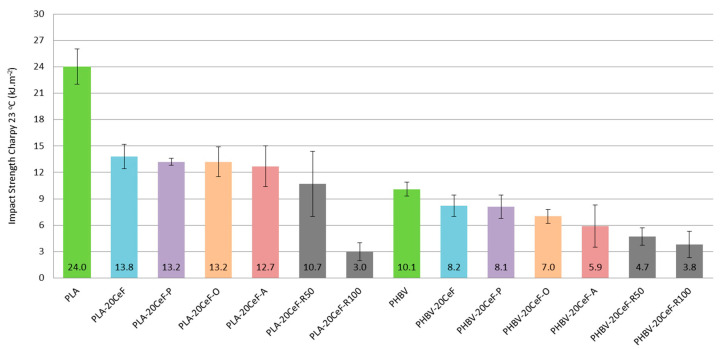
Impact strength Charpy of PLA and PHBV biopolymer composites.

**Figure 13 polymers-12-03006-f013:**
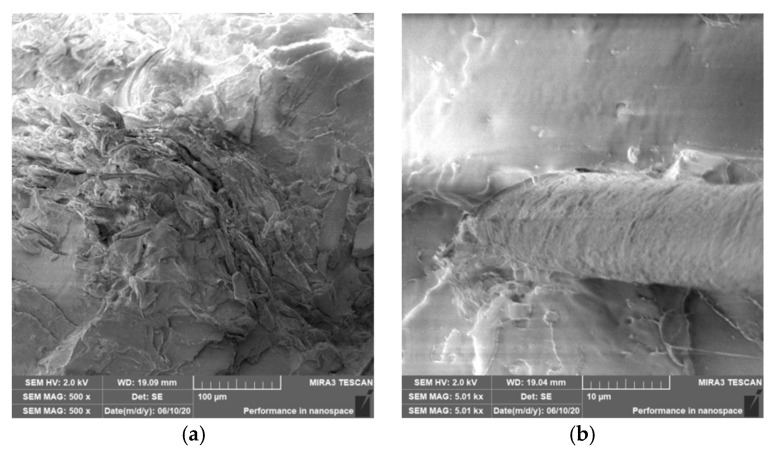
Microscopic image of fracture surfaces of biocomposite structure with the PLA matrix and cellulose fibers (CeFs) without modification of the fiber surface: fracture surface (**a**) and detail of the fiber–matrix interface (**b**), SEM.

**Figure 14 polymers-12-03006-f014:**
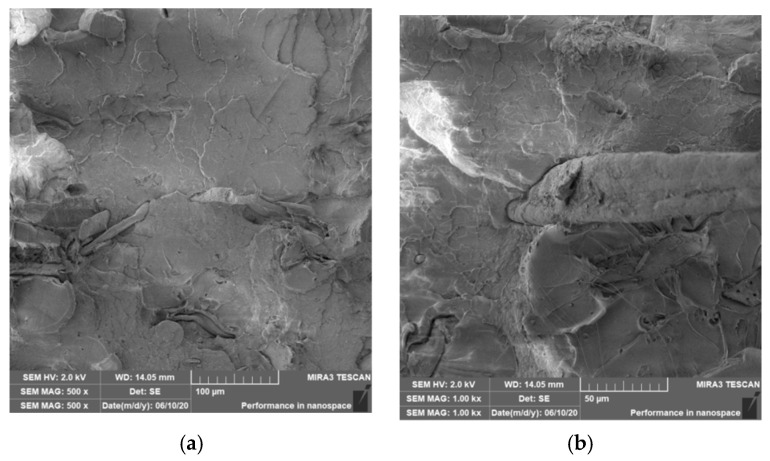
Microscopic image of fracture surfaces of the biocomposite structure with the PLA matrix and cellulose fibers (CeFs) with plasma surface modification of fibers: fracture surface (**a**) and detail of the fiber–matrix interface (**b**).

**Figure 15 polymers-12-03006-f015:**
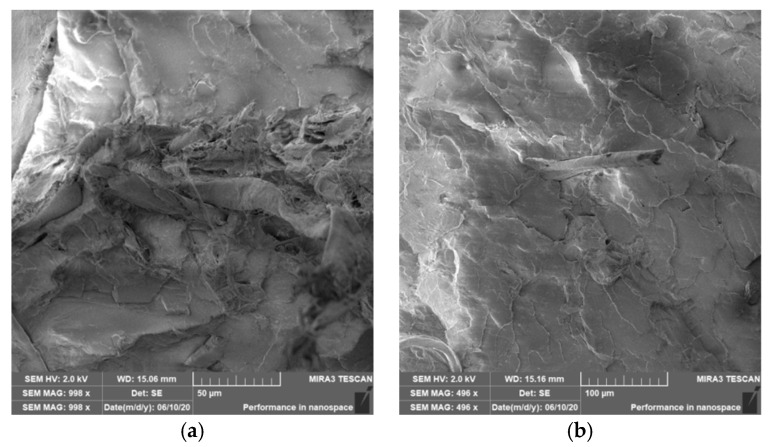
Microscopic image of fracture surfaces of a biocomposite structure with the PLA matrix and cellulose fibers (CeFs) with modification of the fiber surface by ozonation: fracture surface (**a**) and detail of the fiber–matrix interface (**b**).

**Figure 16 polymers-12-03006-f016:**
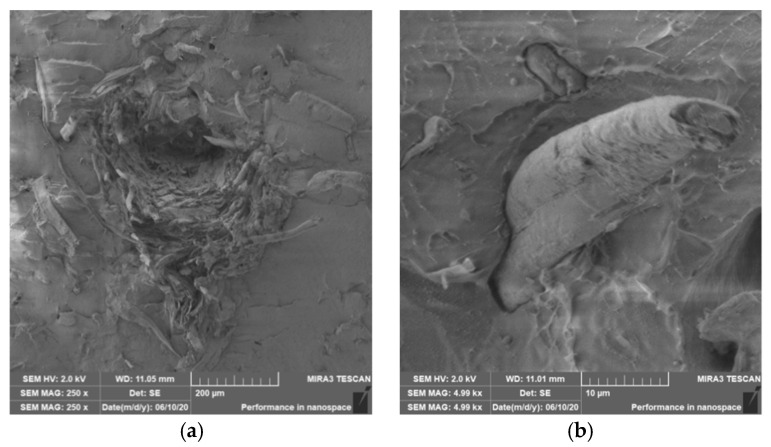
Microscopic image of fracture surfaces of the biocomposite structure with the PLA matrix and cellulose fibers with modification of the fiber surface by acetylation: fracture surface (**a**) and detail of the fiber–matrix interface (**b**).

**Figure 17 polymers-12-03006-f017:**
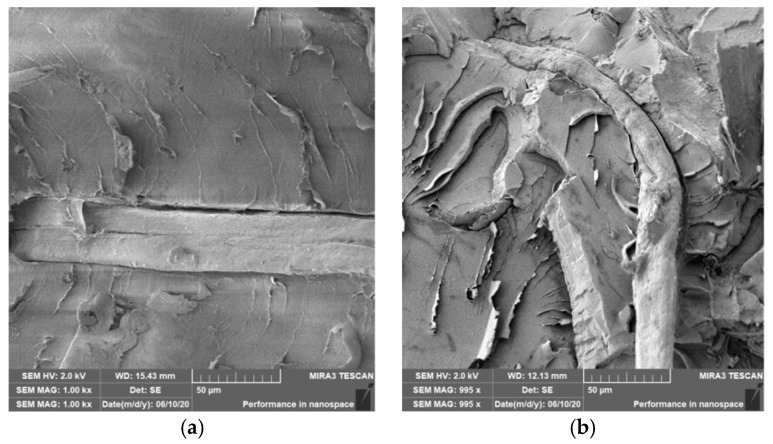
Microscopic image of fracture surfaces of the biocomposite structure with the PLA matrix and cellulose fibers (CeFs) after radiation crosslinking: 50 kGy (**a**) and 100 kGy (**b**), SEM.

**Figure 18 polymers-12-03006-f018:**
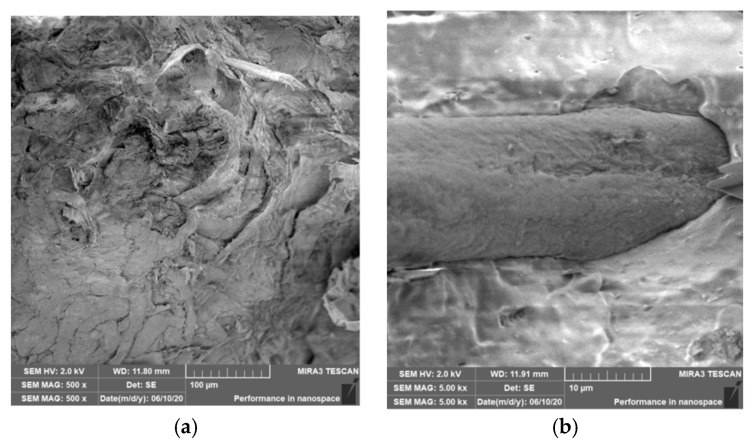
Microscopic image of fracture surfaces of the biocomposite structure with the PHBV matrix and cellulose fibers (CeFs) without modification of the fiber surface: fracture surface (**a**) and detail of the fiber–matrix interface (**b**), SEM.

**Figure 19 polymers-12-03006-f019:**
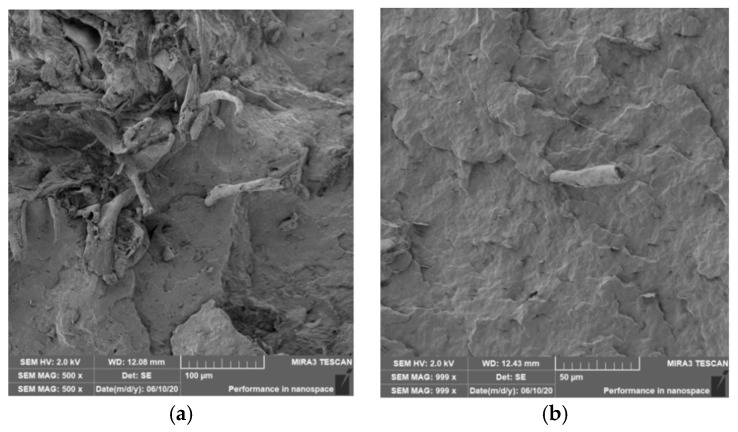
Microscopic image of fracture surfaces of biocomposite structure with the PHBV matrix and cellulose fibers (CeFs) with the plasma surface modification of fibers: fracture surface (**a**) and detail of the fiber–matrix interface (**b**).

**Figure 20 polymers-12-03006-f020:**
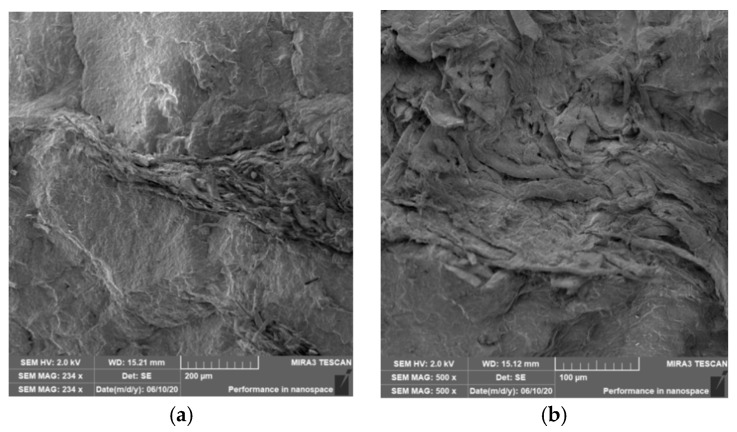
Microscopic image of fracture surfaces of the biocomposite structure with the PHBV matrix and cellulose fibers (CeFs) with modification of the fiber surface by ozonation: fracture surface (**a**) and detail of the fiber–matrix interface (**b**).

**Figure 21 polymers-12-03006-f021:**
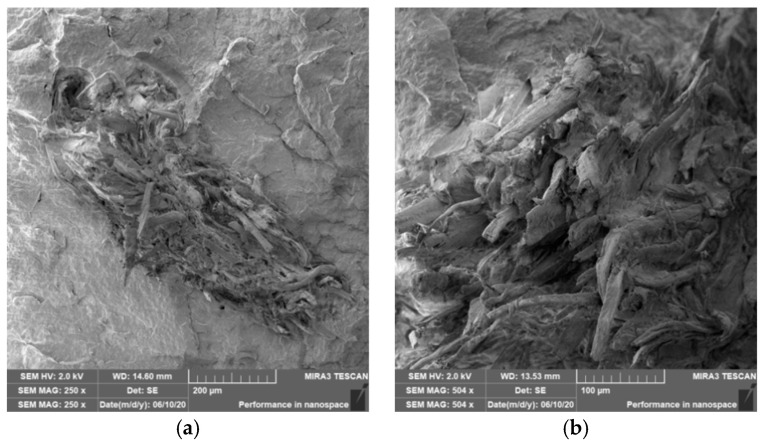
Microscopic image of fracture surfaces of the biocomposite structure with the PHBV matrix and cellulose fibers with modification of the fiber surface by acetylation: fracture surface (**a**) and detail of the fiber–matrix interface (**b**).

**Figure 22 polymers-12-03006-f022:**
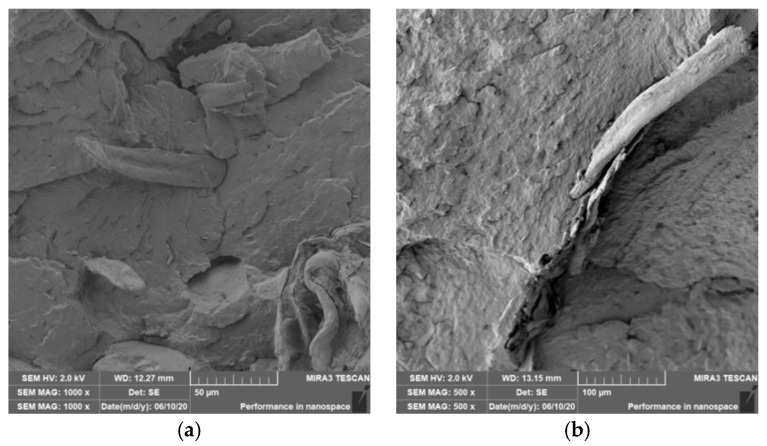
Microscopic image of fracture surfaces of the biocomposite structure with the PHBV matrix and cellulose fibers (CeFs) after radiation crosslinking: 50 kGy (**a**) and 100 kGy (**b**), SEM.

**Table 1 polymers-12-03006-t001:** PLA and PHBV biopolymer composites samples.

Sample Code	Type of Biopolymer	Composition (wt%)		Modification of Fibers	Modification of Composite
		Biopolymer	Cellulose Fibers		
PLA	PLA	100	0	No	No
PLA-20CeF	PLA	80	20	No	No
PLA-20CeF-P	PLA	80	20	Plasma	No
PLA-20CeF-O	PLA	80	20	Ozone	No
PLA-20CeF-A	PLA	80	20	Acetylation	No
PLA-20CeF-R50	PLA	80	20	No	Radiation
PLA-20CeF-R100	PLA	80	20	No	Radiation
PHBV	PHBV	100	0	No	No
PHBV-20CeF	PHBV	80	20	No	No
PHBV-20CeF-P	PHBV	80	20	Plasma	No
PHBV-20CeF-O	PHBV	80	20	Ozone	No
PHBV-20CeF-A	PHBV	80	20	Acetylation	No
PHBV-20CeF-R50	PHBV	80	20	No	Radiation
PHBV-20CeF-R100	PHBV	80	20	No	Radiation

**Table 2 polymers-12-03006-t002:** Differential scanning calorimetry (DSC) results of PLA biopolymer composites samples.

Sample Code (First Heating)	Temperature of Cold Crystallization (°C) T_p,cc_	Enthalpy of Cold Crystallization (J/g) ΔH_cc_	Temperature of Premelt Crystallization (°C) T_p,pc_	Enthalpy of Premelt Crystallization (J/g) ΔH_pc_	Melting Temperature (°C) T_p,m_	Enthalpy of Melting (J/g) ΔH_m_	Degree of Crystallinity (%) X_c_
PLA	103.2	−30.30	156.7	−0.81	171.7	40.74	10.3
PLA-20CeF	97.4	−27.52	155.7	−2.54	171.7	41.74	15.7
PLA-20CeF-P	97.7	−25.60	155.9	−2.44	171.3	41.70	18.3
PLA-20CeF-O	97.5	−28.16	155.9	−2.56	171.7	40.67	13.4
PLA-20CeF-A	96.8	−26.43	154.2	−2.41	169.1	39.09	13.8
PLA-20CeF-R50	98.6	−28.58	154.3	−1.25	167.6	42.92	17.6
PLA-20CeF-R100	98.9	−32.10			165.9	43.67	15.5

**Table 3 polymers-12-03006-t003:** Differential scanning calorimetry (DSC) results of PHBV biopolymer composites samples.

Sample Code (First Heating)	Melting Temperature (°C) T_p,m1_	Melting Temperature (°C) T_p,m2_	Enthalpy of Melting (J/g) ΔH_m_	Enthalpy of Melting (J/g) ΔH_m1_	Enthalpy of Melting (J/g) ΔH_m2_	Degree of Crystallinity (%) X_c_
PHBV	177.4	177.4	93.45			63.7
PHBV-20CeF	171.7	174.3	82.48	59.88	22.60	70.3
PHBV-20CeF-P	171.9	174.9	82.85	69.66	13.19	70.6
PHBV-20CeF-O	171.0	174.7	82.00	69.88	12.12	69.9
PHBV-20CeF-A	172.7	177.0	84.95	71.52	13.43	72.4
PHBV-20CeF-R50	169.9	176.1	86.65	71.36	15.29	73.9
PHBV-20CeF-R100	169.3	174.7	82.71	71.51	11.20	70.5

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
