# Peer review of "Effect of Radiation Crosslinking and Surface Modification of Cellulose Fibers on Properties and Characterization of Biopolymer Composites"

_polymers, 2020, doi:10.3390/polym12123006_

Round 1
Reviewer 1 Report
The paper entitled "Effect of Radiation Crosslinking and Surface Modification of Cellulose Fibers on Properties and Characterization of Biopolymer Composites" by Lenfeld and co. is describing the evaluation of the influence and effect of the type of surface modification of cellulose fibers using physical methods (low-temperature plasma, ozone application, gamma radiation) and chemical methods (acetylation) on the final properties of biopolymer composites.
The paper presents interesting results but some observations must be taken in account, as follows:
1) pag 9 rows 313-319: the authors explained:"Therefore, it can be assumed that a biopolymer with a higher content of crystalline phase is characterized by a higher resistance to radiation and thus a lower decrease in mechanical properties [35]. However, scissions in the linked molecules are the ones that causes the highest losses in mechanical properties to a crystalline polymer [36]. The increase in the tensile modulus of the PHBV biocomposite can be explained by the fact that the cleavage of the amorphous regions was eliminated by increasing the crystal structure"
For the particular PLA and PHVB samples used in the work, it is not clear what are their cristallinity degree values to assume the behaviour from the literature. Some more informations about the cristallinity of all the samples (pristine or treated) must be added (mainly from additonal XRD or even from FTIR measurements).
For plasma treatments some re-crystallization processes can appear due to the high temperature obtained at the polymeric surface.
Depending on the particular composition and crystal structure of the used samples, the degradation of the samples can appear at some input parameneters of the treatments, affecting also the cristallyne regions and not only the amorphous ones, so a decrease in the cristallinity degree can be expected. Moreover, the authors explained a high rate of degradation of PLA samples at page 10 rows 359-360 " In PLA, there is a significant degradation of the polymer matrix due to its chemical composition. "
2) pages 20,21 Conclusions: the obserations from rows 732-796 bust be inserted in the "Results and discussions" part
Author Response
Point 1: For the particular PLA and PHVB samples used in the work, it is not clear what are their cristallinity degree values to assume the behaviour from the literature. Some more informations about the cristallinity of all the samples (pristine or treated) must be added (mainly from additonal XRD or even from FTIR measurements).
Response 1:The crystallinity values and related enthalpies/temperatures for biopolymer composites were added. Measurements were performed by differential scanning calorimetry (DSC). See L325-333 and Table 2 and Table 3.
Point 2: For plasma treatments some re-crystallization processes can appear due to the high temperature obtained at the polymeric surface.
Response 2: During the preparation of the samples were plasma modified only cellulose fibers (not whole biocomposite) so that recrystallization of the polymer could not occur. Low-temperature plasma (DBD) was used for the plasma modification of the fibers precisely in order to avoid thermal degradation of the fibers.
Point 3: pages 20,21 Conclusions: the obserations from rows 732-796 bust be inserted in the "Results and discussions" part.
Response 3: The text was based on results mentioned in the chapter "Results and discussions". Removed.

Reviewer 2 Report
In the present work, the authors have performed a careful investigation on the radiation crosslinking and surface modification of Cellulose fibers. The effects on mechanical properties and characterization of Biopolymer composites were explored. They have presented many experiments to prepare the material, characterize the material, and test material properties. The results are credible, and the language sounds good. The logic of the paper is well organized. I can recommend its publication in the esteemed journal of Polymers.
However, several questions remain which should be carefully treated.
- Please check the sentence of L64. “result in to” should be “result in”.
- The author mentioned that the Lignocellulosic fibers are hydrophilic. How to judge it?
- The plasma technique can be used to treat solid surfaces, such as rock surface. It causes roughness and chemical variation. Please see the recent work: Yu H., Gong L. K., Qu Z. Y., Hao P., Liu J. L., Fu L. Y., Wettability enhancement of hydrophobic artificial sandstones by using the pulsed microwave plasma jet. Colloid and Interface Science Communications, 2020, 36: 100266.
- In L244, “module of tensile elasticity” should be “Young’s modulus”.
- The adhesion between two materials should be characterized the work of adhesion. The authors can make some discussion or mention this issue. Please refer to: Liu J. L., Theoretical analysis on capillary adhesion of microsized plates with a substrate. Acta Mechanica Sinica, 2010, 26(2): 217–223.
- Please check “lie in” in L310.
- The authors have presented many results on the mechanical parameters. Actually Fig. 4 and Table 2 are repeated. The authors also need to analyze the data variation for the parameter values.
- What is the definition of the “flexural modulus”? From the dimension, it is not the same as the bending stiffness.
- The authors can present several formulas or equation if possible.
- The conclusion is too long. Please simplify it.
Author Response
Point 1: Please check the sentence of L64. “result in to” should be “result in”.
Response 1:The text has been corrected (L63). Colored yellow.
Point 2: The author mentioned that the Lignocellulosic fibers are hydrophilic. How to judge it?
Response 2: All cellulose based fibers are hydrophilic due to the presence of -OH groups at their surfaces. This is common knowledge. The change in hydrophilicity could be performed by through the measurement of the contact angle between a solid surface and a droplet of liquid on that surface.
Point 3: The plasma technique can be used to treat solid surfaces, such as rock surface. It causes roughness and chemical variation. Please see the recent work: Yu H., Gong L. K., Qu Z. Y., Hao P., Liu J. L., Fu L. Y., Wettability enhancement of hydrophobic artificial sandstones by using the pulsed microwave plasma jet. Colloid and Interface Science Communications, 2020, 36: 100266.
Response 3: During the sample preparation were plasma modified only cellulose fibers. Improved compatibility at the matrix-filler interface was evaluated at SEM observations of impact fractured surfaces. By plasma modification, various functional groups can be added to the surface of natural fibers. The surface roughness of the fibers may change, which may lead to improved mechanical properties. Basic information is given in the article.
Point 4: In L244, “module of tensile elasticity” should be “Young’s modulus”.
Response 4: "Young's modulus", highlighted in yellow (L243) has been added.
Point 6: Please check “lie in” in L310.
Response 6: The text was changed. See L334-335.
Point 7: The authors have presented many results on the mechanical parameters. Actually Fig. 4 and Table 2 are repeated. The authors also need to analyze the data variation for the parameter values.
Response 7: We agree with the comment, the tables have been removed from the article.
Point 8: What is the definition of the “flexural modulus”? From the dimension, it is not the same as the bending stiffness.
Response 8: The equation and text was added (L256-260). Colored yellow.
Point 10: The conclusion is too long. Please simplify it.
Response 10: Conclusion was shortened.

Round 2
Reviewer 1 Report
The paper was revised and can be accepted in present form